# Dominant Factors Affecting Rheological Properties of Cellulose Derivatives Forming Thermotropic Cholesteric Liquid Crystals with Visible Reflection

**DOI:** 10.3390/ijms24054269

**Published:** 2023-02-21

**Authors:** Yuki Ogiwara, Naoto Iwata, Seiichi Furumi

**Affiliations:** Department of Chemistry, Graduate School of Science, Tokyo University of Science, 1-3 Kagurazaka, Shinjuku, Tokyo 162-8601, Japan

**Keywords:** hydroxypropyl cellulose, cholesteric liquid crystals, thermotropic, rheology

## Abstract

Hydroxypropyl cellulose (HPC) derivatives with alkanoyl side chains are known to form thermotropic cholesteric liquid crystals (CLCs) with visible reflection. Although the widely investigated CLCs are requisite for tedious syntheses of chiral and mesogenic compounds from precious petroleum resources, the HPC derivatives easily prepared from biomass resources would contribute to the realization of environment-friendly CLC devices. In this study, we report the linear rheological behavior of thermotropic CLCs of HPC derivatives possessing alkanoyl side chains of different lengths. In addition, the HPC derivatives have been synthesized by the complete esterification of hydroxy groups in HPC. The master curves of these HPC derivatives were almost identical at reference temperatures, with their light reflection at 405 nm. The relaxation peaks appeared at an angular frequency of ~10^2^ rad/s, suggesting the motion of the CLC helical axis. Moreover, the dominant factors affecting the rheological properties of HPC derivatives were strongly dependent on the CLC helical structures. Further, this study provides one of the most promising fabrication strategies for the highly oriented CLC helix by shearing force, which is indispensable to the development of advanced photonic devices with eco-friendliness.

## 1. Introduction

Considering the concern about the mass consumption of finite petroleum resources remaining on the Earth, it is essential to produce functional materials from biomass resources. Additionally, cellulose and its derivatives are attracting interest because of their ability to meet such a demand for the following three reasons: First, cellulose is the most abundant natural polymer on Earth. Thus, cellulose and its derivatives are regarded as sustainable materials with low impact on the environment and human health, unlike those made from petroleum resources. Secondly, cellulose and its derivatives have excellent thermal and mechanical properties. Finally, cellulose derivatives are capable of showing the liquid crystal phase in solutions [1] or suspensions [2]. The liquid crystal phase is dependent on the concentration of cellulose derivatives. Such a liquid crystal phase is called the lyotropic liquid crystalline phase and is often observed for aqueous suspensions of cellulose nanofibers and cellulose nanocrystals [1,2]. Owing to these unique properties of cellulose and its derivatives, numerous studies have been conducted aiming for the application of cellulose and its derivatives to functional materials for a sustainable society.

Hydroxypropyl cellulose (Figure 1, HPC) is one of the cellulose derivatives and is well known to exhibit a liquid crystal phase. When HPC is dissolved in water at a high concentration, the cholesteric liquid crystal (CLC) phase with visible reflection appears [1]. The reflection properties are dependent on the concentration of HPC. As another intrinsic feature, HPC can exhibit a CLC phase depending on the temperature when the hydroxy groups in its side chains are chemically modified through ester bonds [3,4,5], ether bonds [6,7,8], and urethane bonds [9,10]. The liquid crystal phase is known as the thermotropic CLC phase. However, there have been many reports on the CLC of HPC derivatives with alkanoyl side chains, that is, esterified HPC derivatives [3,4,5]. Due to the property of CLC to reflect the light of specific wavelengths depending on temperature as well as the kind of alkanoyl groups [5], the HPC derivatives are expected to apply to versatile photonic devices such as tunable lasers [11,12] and refractive color displays [13,14]. These devices are promising as one of the candidates for replacing conventional petroleum-based photonic devices, as they are made from cellulose derivatives.

The mechanism of the Bragg reflection color change of HPC derivatives with temperature or the length of alkanoyl groups can be explained by the CLC structure. At the CLC phase, rod-shaped CLC molecules self-assemble to form the helicoidal molecular structure, composed of layers in which the molecules orient in a certain direction [15]. These layers are piled parallel to the surface, that is, in planar orientation, as the direction of the orientation vector in each layer is gradually twisted in a clockwise or counterclockwise manner. Almost all HPC derivatives are known to form right-handed CLC structures. This structure causes periodic modulation of the refractive index of the CLC, thereby leading to the emergence of light reflection. The light reflection phenomenon is regarded as a sort of Bragg reflection. The reflection wavelength of CLC (*λ*_ref_) is approximately calculated as follows:(1)$$\lambda_{\mathrm{ref}}=n_{\mathrm{av}}p$$
where *n*_av_ is the average refractive index and *p* is the helical pitch length, that is, the distance between layers until the orientation vectors rotate 360 degrees. However, the *p* value can be controlled by temperature or by changing the length of alkanoyl groups in the side chains of HPC derivatives, *λ*_ref_ can be tuned according to Equation (1). Specifically, the increase in *λ*_ref_, which corresponds to the increase in *p*, is caused by raising the temperature or introducing bulky functional groups to the side chains of HPC derivatives [5,10]. This tunability of *λ*_ref_ enables the manufacture of CLC materials that reflect light of a certain wavelength at a certain temperature by the introduction of appropriate alkanoyl groups on the side chains. Thus, HPC derivatives can be regarded as promising CLC materials for fabricating new sustainable photonic devices from biomass resources.

In recent years, there have been a lot of reports on the utilization of HPC as functional photonic materials with unique features, such as pressure sensors [16], 3D printing inks [17,18], and temperature sensors [19]. In many precedents, an aqueous HPC solution was used as a lyotropic CLC material. However, it is not practical because the volatilization of the solvents affects their optical and rheological properties. At this point, the HPC ester derivatives show a stably thermotropic CLC phase without the aid of any volatile solvents and would be much more adequate for such applications [20].

In addition, when the CLCs from HPC derivatives are applied to photonic devices, it is of prime importance to evaluate and control not only their optical properties but also their mechanical properties, that is, their rheological properties. However, few reports discuss their mechanical properties from a practical standpoint, even though a lot of reports mainly discuss their optical properties. To date, most of the study of the rheological behavior of CLC has focused on that of lyotropic CLCs derived from its aqueous solutions of HPC [21,22] or that of thermotropic CLCs from HPC derivatives under steady shear flow in a non-linear regime [23]. For example, Asada and Onogi independently measured the apparent shear viscosity of aqueous HPC solutions and found that their flow curves are composed of three regions depending on the shear rate [21,24]. They also found that *λ*_ref_ of HPC solution blue-shifts under shear due to the tilt of the helical axis, not due to the decrease in *p*. Navard and colleagues investigated the molecular weight and shear rate dependence of the apparent viscosity of the thermotropic CLC by using the HPC derivative possessing acetylated side chains [23]. To the best of our knowledge, no reports have been made on the linear rheology of thermotropic CLCs from HPC derivatives, even though it is indispensable to understanding the rheological behavior of CLCs. This is because the linear rheology is basically influenced by the internal structure of the sample, such as the CLC structure, and has excellent reproducibility in the measurements. Recently, we investigated the effect of orientation on the rheological behavior of thermotropic CLCs from HPC derivatives. However, the relationship between their rheological behavior and their structure remained unclear [25,26].

In this study, we investigated the rheological behavior of a series of HPC derivatives possessing acetyl, propionyl, or butyryl side chains. To elucidate the practical rheological behavior of the HPC derivatives, rheological measurements in the linear regime were performed at temperatures where they reflect visible light. We first revealed the angular frequency range where the effect of the orientation state of CLCs on rheological properties is apparent. In this angular frequency range, the relaxation process derived from the CLC structure is dominant. Secondly, we constructed the master curves of the HPC derivatives with the reference temperature where they reflect a blue light of 405 nm and compared their rheological behavior. All the master curves of HPC derivatives were identical regardless of the length of the alkyl side chains, suggesting that the rheological behavior is dependent on the helical pitch length in the CLC structure. Finally, we calculated the activation energies for a molecular motion of the HPC derivatives and proposed a promising mechanism to understand the orientation behavior of CLCs.

## 2. Results and Discussion

### 2.1. Synthesis and Characterization of HPC Ester Derivatives

Figure 1 shows the chemical structures of pristine HPC and its ester derivatives possessing alkanoyl side chains. Three kinds of HPC derivatives were synthesized by esterification of hydroxy groups in pristine HPC with alkanoyl chlorides, such as acetyl chloride, propionyl chloride, and butyryl chloride. The detailed procedures are described in Section 3.2. Hereafter, the HPC ester derivatives possessing acetyl (ethanoyl), propionyl, or butyryl groups are defined as HPC-Et, HPC-Pr, and HPC-Bu, respectively.

After syntheses of the HPC ester derivatives, ^1^H-NMR and FT-IR spectral measurements confirmed that all the hydroxy groups in the side chains of HPC are completely esterified with alkanoyl groups (Figure 2). These chemical analyses have been widely adopted for the characterization of HPC derivatives in many previous studies [27,28,29]. First, the FT-IR spectra were measured before and after the esterification of HPC with alkanoyl chlorides. Before the reaction, pristine HPC showed a broad peak in the wavenumber range between 3100 and 3600 cm^−1^, arising from the O-H stretching vibration of hydroxy groups in HPC (Figure 2, left panel, indicated by the green triangle). After the esterification, the broad O-H peak of pristine HPC around 3100–3600 cm^−1^ thoroughly disappeared, accompanied by the clear appearance of a sharp peak at 1700 cm^−1^ (Figure 2, left panel, indicated by the blue triangle). This sharp peak can be assigned to the C=O stretching vibration of alkanoyl groups in the side chains of HPC derivatives. Therefore, the overall FT-IR results indicated that all hydroxy groups in HPC react with alkanoyl chlorides to form ester linkages.

In another chemical analysis, the NMR spectra of HPC-Et, HPC-Pr, and HPC-Bu were measured to evaluate the stoichiometric reaction rates of hydroxy groups in HPC with alkanoyl chlorides (Figure 2, right panel). Here, the average number of esterified hydroxy groups per HPC monomer unit is defined as the degree of esterification (*DE*). The *DE* values were determined by the following equation using integrated values of the peaks of the ^1^H-NMR spectra of HPC ester derivatives.
(2)$$DE=\frac{A\left(7+6MS\right)}{W-nA}$$
where *A* stands for the integrated value of the peak around 5.00 ppm assigned to methine protons of the propyleneoxy termini modified with alkanoyl groups, *n* is the number of protons in each alkanoyl group, and *W* is the sum of the integrated values of all the peaks derived from the HPC derivatives. The average number of combined propylene oxide groups per HPC monomer unit is defined as the molecular substitution (*MS*). In other words, the *MS* value is equal to the sum of the *x*, *y*, and *z* values, as depicted in Figure 1. From the ^1^H-NMR spectrum of pristine HPC, the *MS* value was determined to be 4.00 according to the analytic procedure reported in a previous study [27]. The detailed numerical derivation of Equation (2) is disclosed in the Appendix A.

Considering the chemical structure of pristine HPC, the maximum value of *DE* is 3.00 because the monomer unit of HPC has three hydroxy groups. In this study, the *DE* values of HPC-Et, HPC-Pr, and HPC-Bu were estimated to be 3.00, 2.98, and 2.98, respectively, as shown in Table 1. These results implied that the hydroxy groups of HPC are almost esterified with alkanoyl chlorides since all the *DE* values are nearly equal to 3.00. There was almost no residual hydroxy group in these HPC ester derivatives.

The weight average molecular weight (*M*_w_) and its distribution with respect to the number average molecular weight (*M*_w_/*M*_n_) of a pristine HPC and its ester derivatives were determined by size exclusion chromatography (SEC) measurements using the polystyrene standards. The details are mentioned in Section 3.2. The results of SEC measurements are also listed in Table 1. The *M*_w_ of each HPC ester derivative became larger than that of pristine HPC while maintaining the almost same *M*_w_/*M*_n_. This implies that the degradation of HPC hardly occurs during the esterification process.

It should be emphasized that the preparation of completely-esterified HPC derivatives is indispensable to a fair comparison of their optical and rheological properties. It has been reported that the reflection wavelength of HPC ester derivatives is remarkably affected by the *DE* value. For example, the reflection peak of HPC-Bu was drastically red-shifted from 400 to 835 nm with the decrease in *DE* value from 2.96 to 2.20, possibly because of residual hydroxy groups in HPC-Bu [30]. Note that this wavelength range between 400 and 835 nm fully covers the visible wavelength region and even the near infrared region. The red-shift of the reflection peak with a decreasing *DE* value was attributed to the increase in helical pitch length (*p*) of the CLC structure. Such a large difference in CLC structure may also affect the rheological behavior of CLCs. This is because the rheological properties of viscoelastic materials are basically influenced by their internal structures. Taking the above-mentioned concern into account, three kinds of the completely-esterified HPC derivatives synthesized in this study are rationally suitable for the investigation of their optical and rheological properties.

### 2.2. Reflection Properties of HPC Ester Derivatives

The HPC ester derivatives used in this study, namely HPC-Et, HPC-Pr, and HPC-Bu, showed a thermotropic CLC phase. This is supported by polarized optical microscope (POM) images and reflection images of their CLC cells upon heating. The CLC cells were fabricated by enclosing the HPC derivatives between two glass substrates at an internal gap distance (*d*_g_) of 0.20 mm. The fabrication procedure of the CLC cell is described in detail in Section 3.3. The POM images showed transmitted light under crossed-Nicols, meaning the emergence of optical birefringence by liquid crystallinity (Appendix A) [31]. In addition, the emergence of the CLC phase was also evident from the visual observation of reflection colors upon changing the temperature (Appendix A).

Previous reports have shown that the reflection peak wavelength of thermotropic CLCs of HPC ester derivatives can be tuned by not only the temperature but also the length of alkanoyl side chains [5,10]. In this study, the transmission spectra of HPC-Et, HPC-Pr, and HPC-Bu were measured as the temperature changed (Figure 3A).

Here, the reflection peak wavelength was determined as the wavelength at which the optical transmittance became minimum. Although no reflection peak was observed in the visible wavelength range for the three kinds of HPC ester derivatives at room temperature, the subsequent heating treatment led to the appearance of the reflection peak in the visible wavelength range above ~350 nm. When HPC-Et was heated to 110 °C, the reflection peak appeared at 385 nm (Figure 3A, upper panel). At 112 °C, HPC-Et showed a reflection peak at 405 nm, whereupon the reflection color was visualized as blue. As the temperature was elevated from 112 to 120 °C, the reflection peak was red-shifted from 405 to 460 nm, arising from the increase in the helical pitch of CLC, which corresponds to *p* in Equation (1).

Similarly, the thermally induced shifting behavior of the reflection peak was also observed for the two other HPC derivatives, that is, HPC-Pr and HPC-Bu. For instance, HPC-Pr exhibited a reflection peak at 375 nm by heating it to 95 °C. As the temperature was elevated to 120 °C, the reflection peak was red-shifted to 520 nm (Figure 3A, middle panel). For comparison, a reflection peak of HPC-Bu appeared at 380 nm by heating at 60 °C, while heating treatment at 120 °C gave rise to the red-shift of the reflection peak to 710 nm in a relatively wide wavelength range (Figure 3A, lower panel).

In all cases, the reflection peak was reversibly blue-shifted as the temperature dropped. Notably, the reflection peak of each HPC derivative was quite unchanged with sufficient reproducibility when measured at the same temperature in the heating or cooling process (Appendix A). For instance, the reflection peaks of HPC-Bu appeared at the same wavelength even after repeating the cycles of heating to 100 °C and cooling to 80 °C five times (Appendix A). From the spectral results, the standard deviations of the reflection wavelength at 80 °C and 100 °C were calculated to be 1.12 nm and 1.15 nm, respectively. Those low standard deviations of the reflection wavelength indicate that the reflection properties of the HPC ester derivatives, by changing the temperature, have excellent reversibility. Such a reversible shift of the reflection peak is ascribed to the typical characteristic of thermotropic CLC.

Based on the overall spectral results, the temperature dependences of the reflection peak wavelengths of HPC-Et, HPC-Pr, and HPC-Bu are summarized in Figure 3B. The reflection wavelengths of all the HPC ester derivatives were consecutively red-shifted with increasing temperatures (Figure 3B, red circles for HPC-Et, blue triangles for HPC-Pr, and black squares for HPC-Bu). The length of alkanoyl side chains was found to have a remarkable influence on the reflection peak shift ranges by temperature as well as the CLC temperature ranges with visible reflection. For example, the reflection peak of HPC-Bu could be shifted throughout the full visible wavelength range between 380 and 710 nm by sweeping the temperature from 60 to 120 °C. On the other hand, the reflection peak shift ranges by temperature were quite limited for HPC-Et and HPC-Pr. Additionally, the reflection peaks of HPC ester derivatives at the same temperature, such as 110 °C, appeared at longer wavelengths in order of the length of the alkanoyl side chains. In other words, HPC-Et, HPC-Pr, and HPC-Bu exhibited blue reflection color with the wavelength at 405 nm, adjusting CLC temperatures to 112, 100, and 66 °C, respectively (Figure 3B, dotted line). The difference in CLC temperatures probably originated from the steric hindrance on the alkanoyl side chains of HPC ester derivatives, that is, the alkanoyl chain length, implying that the CLC temperature with the same reflection wavelength is lowered as the length of the alkanoyl side chains is longer.

In addition, the isotropic phase transition temperature (*T*_i_) of the HPC ester derivatives decreased as the side chain length increased. The *T*_i_ of HPC-Et was determined to be 155–165 °C by the POM observation under crossed-Nicols during the rise in temperature. In the same manner, the *T*_i_ of HPC-Pr and HPC-Bu were observed at 150–160 °C and 140–145 °C, respectively. These differences in *T*_i_ affect the temperature range over which the time-temperature superposition (TTS) principle can be applicable. For instance, the breakdown of the TTS principle was more likely to occur with increasing the temperature to *T*_i_ as will be discussed in Section 2.5. Therefore, the reference temperature (*T*_r_) for the construction of the master curve was defined as the temperature at which the light of 405 nm was reflected (Figure 3A). This is because it is the lowest temperature at which each HPC derivative reflects visible light.

### 2.3. WAXD Results of HPC Ester Derivatives

Wide-angle X-ray diffraction (WAXD) measurements give invaluable investigation on the microscopic parameters of CLC structure, thereby leading to the twisting angle (*φ*) and the layer distance (*d*) in helical molecular assemblages, which are defined as the difference in azimuthal angle and distance between the neighboring nematic liquid crystalline layers in the CLC helical structures, respectively. The previous studies on the HPC ester derivatives have revealed that the *d* value increases while the *φ* value decreases, elongating the length of alkanoyl side chains [32]. Thus, the relationship among the physical parameters of *λ*_ref_, *d*, and *φ* can be determined by combining Equation (1) with Equation (3).
(3)$$p=\frac{360}{\varphi }d$$

As derived from the equations, Equation (4) indicates that *λ*_ref_ is proportional to *n*_av_, *d*, and *φ*^−1^ as follows.
(4)$$\lambda_{\mathrm{ref}}=\frac{360\;n_{\mathrm{av}}d}{\varphi }$$

First, the microscopic CLC structure can be determined by measuring the *d* value from WAXD measurements. According to the previous study, the *n*_av_ values of the HPC derivatives are approximately 1.465, regardless of their chemical structures [32]. Strictly speaking, the *n*_av_ of each HPC derivative has been reported as 1.468 for HPC-Et, 1.465 for HPC-Pr, and 1.464 for HPC-Bu. However, the *n*_av_ value can be considered a constant since these differences are quite small. Therefore, they have the same *p* value as 276 nm when they reflect lights at 405 nm according to Equation (1). Furthermore, the WAXD measurements indicated that the number of layers of HPC derivatives is almost the same when they reflect light at a wavelength of 405 nm (Appendix A). We assigned the peaks appearing around 2*θ* = 6° and halos appearing around 2*θ* = 20° to the distance between piled layers and nematic structure in a layer, respectively. These peak assignments of WAXD profiles were in good agreement with those of previous studies [32,33,34]. The parameters of the CLC structure determined by WAXD measurements are summarized in Table 2.

The *d* values were calculated to be 1.23 nm for HPC-Et, 1.29 nm for HPC-Pr, and 1.33 nm for HPC-Bu from their 2*θ* values using Equation (5) for WAXD measurements.
(5)$$d=\frac{\lambda }{2\sin\theta }$$
where *λ* means 0.154 nm as the wavelength of Cu-Kα radiation. This small increase in *d* value is reasonable due to the increase in the alkanoyl side chain length of HPC ester derivatives. The *φ* values were obtained by using Equation (4). We defined *N*_Layer_ as the number of layers in a periodic helical structure calculated by both Equations (1) and (6).
(6)$$N_{\mathrm{Layer}}=\frac{p}{d}=\frac{\lambda_{\mathrm{ref}}}{n_{\mathrm{av}}d}$$
where the *n*_av_ value is 1.465 and *λ*_ref_ is 405 nm. Therefore, the *p* value is estimated to be 276 nm. From Equation (6), the *N*_Layer_ values were found to be 224 for HPC-Et, 214 for HPC-Pr, and 208 for HPC-Bu, so that the HPC ester derivatives exhibited almost the same *N*_Layer_ values. Taking the overall results into account, it is plausible that three kinds of HPC ester derivatives form almost the same CLC structures at the temperatures where they reflect light at 405 nm. As a result, HPC-Et, HPC-Pr, and HPC-Bu are suitable for a fair comparison of their rheological properties. For this reason, we determined *T*_r_ for the construction of master curves as the temperature at which HPC derivatives reflect a light of 405 nm in the following rheological measurements.

### 2.4. Rheological Properties of HPC Ester Derivatives: Gap Dependence of Storage and Loss Moduli

As preliminary experiments, we elucidated the dependence of the rheological properties of HPC-Pr on the gap distance (*d*_g_), which corresponds to its thickness between the parallel plate and sample stage in the rheometer, as stated later in Section 3.5. These experiments are necessary to examine the rheological behavior of HPC derivatives because the rheological properties of liquid crystalline polymers are greatly affected by differences in their internal structure, such as molecular orientation or domain size in general. Figure 4 shows the changes in storage modulus (*G*′) and loss modulus (*G*″) of HPC-Pr as a function of the angular frequency (*ω*). The rheological measurements were conducted in the same conditions except for the *d*_g_ value, which was adjusted to 0.20, 0.25, 0.30, 0.75, and 1.00 mm. As mentioned in Section 3.5, the strain amplitude was tuned at 0.7%. It should be noted that the strain amplitude of 0.7% is small enough to measure the linear viscoelasticity regardless of the *d*_g_ value. This ensures that the rheological behavior of HPC derivatives is only governed by the difference in CLC structures.

As shown in Figure 4, both *G*′ and *G*″ values of HPC-Pr were distinctly affected by the *d*_g_ value in the rheological measurements. However, the values of *G*′ and *G*″ were almost identical at the *d*_g_ values of 0.75 and 1.00 mm. Theoretically, the *G*′ and *G*″ values are considered independent of the *d*_g_ value in the linear rheology. In fact, the rheological profiles overlapped at *d*_g_ values between 0.75 and 1.00 mm. At this *d*_g_ range, the values of *G*″ were constantly higher than those of *G*′ in the entire *ω* range. Furthermore, it turned out that both *G*′ and *G*″ values continuously decreased by reducing the *ω* value. For example, at *d*_g_ = 0.75 and 1.00 mm, as *ω* dropped from 10^2^ to 10^−1^ rad/s, the *G*′ value fell from 2.0 × 10^2^ to 4.0 Pa and the *G*″ value also decreased from 1.8 × 10^3^ to 6.0 Pa. These two trends suggest the liquid-like behavior of CLC. However, in the region below *ω* = 10^1^ rad/s, the slopes of both *G*′ and *G*″ values against *ω* were nearly equal to 0.60 and 0.75, respectively. This relationship is different from those of the ideal Newtonian fluid, which would be $G^{\prime }\propto \omega^{2}$ and $G^{\prime \prime }\propto \omega^{1}$, respectively. Such a weaker *ω* dependence of the *G*′ and *G*″ values can be attributed to the disturbance of the flow behavior caused by the CLC structure, since a similar *ω* dependence can also be found for the polymer nanocomposites containing an aggregation of fillers [35].

Additionally, both *G*′ and *G*″ values increased drastically when *d*_g_ was decreasingly adjusted from 0.30 mm to 0.20 mm, and the slopes of *G*′ and *G*″ values gradually decreased to 0. When *d*_g_ was set at 0.25 mm, both *G*′ and *G*″ values at *ω* = 10^−1^ rad/s were almost equal, approximately 20 Pa and 18 Pa, respectively (Figure 4, blue plots). In the *ω* range from 10^−1^–10^0^ rad/s, both *G*′ and *G*″ values were three times larger than those of *d*_g_ = 0.75 and 1.00 mm. The slopes also became more gradual; specifically, those of *G*′ and *G*″ in the *ω* range below 10^0^ rad/s were approximately 0.23 and 0.59, respectively. Furthermore, the plots of *G*′ and *G*″ were distinct from the others when the gap was further thinned to 0.20 mm. The *G*′ and *G*″ values in the *ω* range below 10^1^ rad/s became independent of *ω*, and their values were approximately 250 Pa and 280 Pa, respectively, which are 60 or 45 times larger than those of *d*_g_ = 0.75 and 1.00 mm. The increase of shear moduli (*G*′ and *G*″) and decrease of *ω* dependency indicated that the CLC showed solid-like behavior. This tendency was especially pronounced in the low *ω* region, particularly below *ω* = 10^1^ rad/s. Such a transition in the viscoelastic properties of HPC-Pr depending on the *d*_g_ value might be caused by the orientation of CLC molecules. Since the decrease in the *d*_g_ value enhances the effect of surface anchoring from the jig interfaces of the rheometer, the HPC derivatives form more rigid structures due to the strongly planar CLC orientation, leading to the solid-like behavior of HPC-Pr. The effect of the *d*_g_ value on the CLC structure is called the wall effect in a previous study [24].

In addition, this hypothesis can also be supported by the difference in transmission spectra of HPC-Pr with various *d*_g_ values. The transmission spectra of CLC cells, fabricated by enclosing HPC-Pr with two glass substrates in the *d*_g_ range of 0.10–1.00 mm, were measured at 120 °C (Appendix A). As the *d*_g_ increased, the baselines of the transmission spectrum were gradually lowered, accompanied by a broadening in the spectral widths of the reflection peaks. These results were probably caused by the light scattering due to the disordered CLC structure induced by thickening the gap. The differences in reflection peak wavelength between the CLC cells with different *d*_g_ values are probably due to the broadening of the reflection peaks. This is reasonable because the reflection wavelength is greatly affected by the enlargement of the distribution of *p* accompanied by the increase in *d*_g_ as evident from Equation (1). Furthermore, the enhancement of CLC orientation with decreasing the *d*_g_ value is also supported by the reflection spectral results (Appendix A). We measured the changes in the reflection spectrum of HPC-Bu after applying the steady shear flow. Although the reflection spectra of HPC-Bu became broader and the reflection peaks were blue-shifted right after applying the shear stress, the reflection spectra gradually recovered due to the CLC orientation. According to previous studies, the helical axis of CLC tilts under the shear flowing force, and the oblique helical axis gradually recovers into a planar orientation after cessation of the shear flow [21,36]. Thus, we can estimate the CLC orientation behavior by measuring the time needed for the complete recovery in the reflection spectrum. At this time, the *d*_g_ value was set to be either 0.2 or 1.0 mm to reveal the effect of the thickness of CLC. After applying the shear stress, the reflection spectrum was recorded every 10 min until the complete recovery of the reflection spectrum using our original rheo-optics measurement system (Appendix A). The result suggested that the relaxation time after the shear flow is shorter when the gap is smaller, from 105 min at *d*_g_ = 1.0 mm (Appendix A) to 65 min at *d*_g_ = 0.20 mm (Appendix A). The fast recovery of the reflection spectrum when *d*_g_ = 0.20 mm strongly suggests that CLC molecules of HPC ester derivatives are more likely to be oriented when the *d*_g_ value is small.

### 2.5. Rheological Properties of HPC Ester Derivatives: Master Curves

The master curves of HPC-Et, HPC-Pr, and HPC-Bu were almost identical when they were constructed with the *T*_r_ where each derivative reflected a blue light at 405 nm (Figure 5A, red plots for HPC-Et, blue plots for HPC-Pr, and black plots for HPC-Bu). The *T*_r_ of HPC-Et, HPC-Pr, and HPC-Bu are 112 °C, 100 °C, and 66 °C, respectively. These *T*_r_ values were determined from the results of optical measurements to ensure that all the HPC derivatives have the same CLC structure for a fair comparison of their rheological properties, as mentioned in Section 2.3. To make master curves with a wide range of *ω*, *ω* dependence of *G*′ and *G*″ in the range between 10^−1^ and 10^2^ rad/s was measured at different temperatures and the results were combined according to the TTS principle. The TTS principle was applicable to the results of *G*′ and *G*″ in the temperature ranges below 112 °C for HPC-Et, 100 °C for HPC-Pr, and 80 °C for HPC-Bu, even though they exhibited the CLC phase. The validity of the TTS principle is also supported by the van Gurp-Palmen (vGP) plot, which can be obtained by plotting the phase angle (*δ*) against the corresponding absolute value of the complex shear modulus (*G**) [37].

The vGP plots of HPC derivatives are presented in Appendix A. Each curve obtained at different temperatures was overlapped to show a smooth curve, which is a proof for the validity of the TTS principle. However, the TTS principle was not applicable for the measurements at temperature ranges higher than the above-mentioned temperatures. This is probably due to the gradual transition to the isotropic phase, which may somehow destroy the CLC structure. For instance, the vGP plots of HPC-Et were well overlapped in the temperature range between 15 °C and 112 °C, but they were not overlapped when the temperature was elevated to 140 °C. When the temperature was further increased to 150 °C, *δ* became nearly equal to 90° regardless of *G** (Appendix A, upper panel). The *δ* of 90° suggests that viscoelastic behavior is dominated only by the viscosity while the sample behaves as a fluid. Such fluid-like behavior was probably caused by the disappearance of the CLC structure due to the phase transition from the CLC phase to the isotropic phase. In fact, the isotropic phase transition temperature (*T*_i_) of HPC-Et was determined to be 155–165 °C, as described in Section 2.2. In the same manner, the *T*_i_ of HPC-Pr and HPC-Bu were observed at 150–160 °C and 140–145 °C, respectively. These temperature ranges were nearly equal to those at which the breakdown of the TTS principle, that is, discontinuity of vGP plots, was observed (Appendix A, middle and bottom panels). From these results, we concluded that the TTS principle can be applicable for the construction of the master curves of HPC derivatives below their *T*_i_s.

The fluid-like behavior of HPC-Pr above *T*_i_ can also be confirmed from its temperature dependence and frequency dependence of *G*′ and *G*″. The temperature dependence of HPC-Pr is given in Appendix A. In this profile, a peak caused by the isotropic phase transition was observed at 150 °C, which was very close to the *T*_i_ of HPC-Pr determined by POM observation, that is, 150–160 °C. Above this temperature, *G*′ and *G*″ rapidly decreased due to the fluid-like behavior of CLC in the isotropic phase. This tendency was consistent with the vGP plots shown in Appendix A. The frequency dependence at 155 °C shown in Appendix A suggests that HPC-Pr is flowing since the slopes of *G*′ and *G*″ are constant at 1.0. This indicates that the CLC structure seems to hinder the Newtonian flow of the molecules even at the isotropic phase. Such flow behavior of HPC-Pr above its *T*_i_ supports the deformation of the CLC structure, which induces the breakdown of the TTS principle.

The master curves of HPC derivatives were almost identical, even though they were constructed with widely different *T*_r_. This result strongly suggests that the rheological behavior of HPC derivatives probably depends on the reflection wavelength of the CLCs, corresponding to *p*. The reasons for such identification will be discussed later.

As highlighted in yellow in Figure 5A, the higher *ω* region over 10^5^ rad/s is considered to be a glass region. The intersections of *G*′ and *G*″ appeared around *ω* = 10^5^–10^6^ rad/s for all HPC derivatives (Figure 5A, green arrows). Since the *G*″ is always greater than *G*′ on the lower *ω* side of these intersections, it can be assumed that this drastic change in rheological properties may be related to the glass transition of HPC derivatives. In this region, the relaxation processes due to the thermal motion of side chains or the micro-Brownian motion of the main chain seem to be dominant, similar to those of amorphous polymers (Figure 5B). Therefore, we synthesized HPC-Pr with different molecular weights (Appendix A). Their master curves indicate that the micro-Brownian motion of the main chain affects their rheological phenomena in the glass region (Appendix A). As *M*_w_ increased, the master curves of these HPC derivatives in the glass region tended to broaden, though their values of glass transition temperature (*T*_g_) remained constant at −23 °C regardless of *M*_w_. This tendency implies that the micro-Brownian motion from the main chain contributes to rheological behavior in the higher *ω* region above 10^5^ rad/s. However, side-chain motion seems to have a greater influence on the viscoelastic behavior in the glass region. As the side chain length of HPC derivatives increased, the *T*_g_ values were significantly lowered; for instance, −8 °C for HPC-Et, −23 °C for HPC-Pr, and −37 °C for HPC-Bu, as determined by temperature-dependent rheology measurements conducted at 1.0 Hz (Appendix A). Although the difference in the side chain length of HPC derivatives is relatively small because the side chain of HPC consists of 1–2 hydroxypropyl groups and alkanoyl groups. However, this small difference in side chain structure dramatically affects *T*_g_, the influence of the side chain on the glass region seems to be dominant for the difference in this relaxation.

The master curves of each derivative were consistent in this high *ω* region over 10^5^ rad/s, although the side chain lengths and the *T*_r_ values were different. This was confirmed by the fact that the intersections of *G*′ and *G*″ appear at approximately equal frequencies; that is, *ω* = 10^5^–10^6^ rad/s (Figure 5A, green arrows). In general, the viscoelastic behavior in the glass region is greatly affected by the structure of the side chain, as mentioned above. Nevertheless, the overlap of the master curves in this study is due to the offsetting effects of side chain length and *T*_r_ values. If the *T*_r_ values are the same, the master curves of the derivatives possessing longer side chains tend to shift to the higher *ω* side when considering *T*_g_ values and the TTS principle. On the other hand, as discussed in Section 2.2, as the length of the side chain increased, the *T*_r_ values, which correspond to the temperature at which the derivative reflects 405 nm blue light, were lowered, and the master curves shifted to the lower *ω* side. As a result of these two opposite shift effects, these master curves seemed to be consistent in the high *ω* region.

On the other hand, all the inflection points of *G*′ appeared at *ω* = 10^2^ rad/s regardless of the length of the alkanoyl side chains (Figure 5A, purple triangles). The appearance of inflection points is probably due to the CLC structure of HPC derivatives, which induces flattering of the slopes of *G*′ and *G*″ on the low *ω* side, as we discussed in Section 2.4. It should be emphasized that this is not because of the entanglement of HPC derivatives since no rubbery plateau appeared in the temperature dependence of *G*′ and *G*″ of HPC-Pr (Appendix A). This indicates that the dominant relaxation process is not due to a molecular motion but to a larger scale motion such as the tilt of the helical structure of CLC (Figure 5C). Since the viscoelastic behavior is independent of the side chain length, the relaxation mechanism due to the tilting motion of the helical axis of the CLC, illustrated in Figure 5C, may be dominant in this *ω* region below 10^2^ rad/s, as highlighted in purple in Figure 5A. A previous study has reported that the helical axis of the CLC tilts when shear is applied to CLCs [21]. Since each layer of the helix is uniformly tilted by shear force, it is expected to show similar viscoelastic behavior regardless of the structure of the side chain.

This proposal about the dominant molecular motion on the low *ω* side is also supported by our previous work [25]. We investigated the effect of orientation state in CLC on rheological behavior. Dynamic viscoelastic measurements of HPC-Pr were conducted after the thermal treatment in isotropic phase or shearing at a constant shear rate of 10 s^−1^ for 200 s. These pre-treatments before the rheological measurements ensured that CLC would have different structures, that is, a randomly-arranged state or a shear-oriented state. As a result of the *ω* dependence of *G*′ and *G*″ in HPC-Pr at 100 °C, the difference was apparent, especially in the *ω* region below 10^1^ rad/s. This result indicates that the effect of the orientation of CLC structure appears on the low *ω* side, which is consistent with the result of this study.

The consistency of master curves for each derivative in the low *ω* region below 10^2^ rad/s is due to the same CLC structures. All the HPC derivatives had almost the same CLC structure when they reflected blue light at 405 nm, as confirmed by optical measurements and WAXD measurements, as stated in Section 2.3. Furthermore, in this *ω* region, the relaxation process caused by the CLC structure was dominant, as described in Section 2.4. Therefore, the similar viscoelastic behavior, regardless of the length of the side chain, can be attributed to the identical structure of the CLCs.

Considering the two characteristics of the master curves mentioned above, the master curves of HPC derivatives can be divided into two regions depending on the *ω*. It can be assumed that the highest *ω* range above 10^5^ rad/s is a glass region at which the molecular motion of CLC molecules is frozen. In the lowest *ω* range below 10^2^ rad/s, large-scale motion such as tilting of the helical axis may be dominant. The intermediate *ω* range, that is, the *ω* range of 10^2^–10^5^ rad/s, can be estimated to be the transition region between the two regions mentioned above. In this region, the motion of the helical axial unit gradually freezes, and the motion of the side chains and main chains becomes dominant with the increase of *ω*.

The horizontal shift factors (*a*_T_) used for the construction of the master curves of HPC derivatives also showed unique behavior. The profiles of *a*_T_ values for HPC-Et, HPC-Pr, and HPC-Bu are shown in Appendix A. All of them decreased logarithmically with raising the temperature to the *T*_r_, meaning a typical characteristic for *a*_T_ of polymers. In addition, the *a*_T_ values were almost the same for all the HPC derivatives synthesized in this study. This result suggests that the temperature dependence of the time scale of molecular motion is not affected by the length of alkanoyl side chains in HPC derivatives. Another interesting property of *a*_T_ of HPC derivatives appeared when they were fitted with the Williams-Landel-Ferry (WLF) equation shown in Equation (7) (Appendix A, solid lines) [38].
(7)$$\log a_{T}=\frac{-c_{1}\left(T-T_{r}\right)}{c_{2}+\left(T-T_{r}\right)}$$

The WLF parameters *c*_1_ and *c*_2_ for HPC derivatives are listed in Appendix A. The values of *c*_1_ and *c*_2_ for all the derivatives were almost identical, and they are in the range of 2.38–2.53 and 126–148 K, respectively, suggesting that the values are also close to those of amorphous polymers. This tendency implies that the *a*_T_ values of the HPC derivatives show WLF-type temperature dependence.

### 2.6. Rheological Properties of HPC Ester Derivatives: Arrhenius Plots of Horizontal Shift Factors

The activation energies of each relaxation process (*E*_a_) were able to be calculated by the Arrhenius-plots of *a*_T_ values, which were used to construct the master curves of HPC-Et, HPC-Pr, and HPC-Bu in Section 2.5 (Figure 6). The complete explanation of the activation energies of the relaxation process is available in the Appendix A. The Arrhenius plot of the *a*_T_ showing the WLF-type temperature dependence is generally a curve that increases as a power function with increasing 1/*T*. Note that *T* is the absolute temperature in the unit of Kelvin [39]. However, when the temperature range is narrow, the Arrhenius plot can be approximated well by a straight line, and the *E*_a_ can be obtained by its slope. This is because *a*_T_ satisfies Equation (8) as follows:(8)$$\ln a_{T}=\frac{E_{a}}{R}\cdot \frac{1}{T}+\mathrm{const}.$$
where *R* is the gas constant. Such a methodology has been adopted in many previous studies [39,40,41].

The value of *E*_a_ here means the magnitude of the energy barrier for the molecules to move. The CLC, which is in an excited state due to the application of strain, returns to the ground state by the molecular motion. In dynamic viscoelasticity measurements, the stress in the relaxation process is determined, so *E*_a_ refers to the size of the energy barrier for the motion of the molecules when the molecules return from the excited state to the ground state. In other words, *E*_a_ can be considered to represent the mobility of CLC in a relaxation mode in the temperature range where linear Arrhenius plots can be yielded. The complete derivation of Equation (8) is also available in the Appendix A.

The Arrhenius plots of the three HPC derivatives investigated in this study are presented in Figure 6. These plots were made by plotting the natural logarithm of *a*_T_ against the inverse of the temperature. As expected, the natural logarithm of *a*_T_ of HPC-Et increased in a power law fashion as the temperature decreased (Figure 6, upper panel). Notice that the increase in 1/*T* corresponds to the decrease in *T*.

The value of *E*_a_ calculated from the slope of the regression line of the Arrhenius plots in higher temperature regions may suggest the ease of orientation in the CLC structure. We fitted them with a regression line, as shown in Figure 6. As mentioned in Section 2.5, the master curves can be divided into two regions, and the rheological behavior in the lower *ω* region, that is, the high-temperature region, is dominated by the relaxation process caused by CLC structure. Considering this result, the *E*_a_ value from the plots in the high-temperature region represents the activation energy derived from the helical structure of CLC. The *E*_a_ value decreased with increasing the alkanoyl side chains of HPC derivatives, that is, 58.4 kJ/mol for HPC-Et, 57.0 kJ/mol for HPC-Pr, and 53.5 kJ/mol for HPC-Bu, as shown in Figure 6. Since the *E*_a_ value means the mobility of molecules, as described in the Appendix A, the molecules become easier to move as the side chain length increases, considering the result that *E*_a_ decreases as the side chain length increases. This may be related to the improvement of the orientation of CLCs.

## 3. Materials and Methods

### 3.1. Materials

HPC (Fujifilm Wako Pure Chemical Co., Osaka, Japan; viscosity of a 2.0 wt% aqueous solution: 2.0–2.9 mPa·s at 20 °C) was used as a starting material for the synthesis of its derivatives. Size-exclusion chromatography (SEC) measurement using the polystyrene standards revealed that the number average molecular weight (*M*_n_) and weight average molecular weight (*M*_w_) are 2.30 × 10^4^ and 4.45 × 10^4^, respectively. The average number of hydroxypropyl groups per HPC monomer unit, that is, the molar substitution value (*MS*), was determined to be 4.00 by measuring the ^1^H-NMR spectrum of pristine HPC in CDCl_3_ according to the previous report [27]. The *MS* value means the sum of *x*, *y*, and *z* in Figure 1. HPC was dried under vacuum at room temperature for over 24 h before use.

Anhydrous acetone (Kanto Chemical Co., Inc., Tokyo, Japan; 99.5%) and anhydrous pyridine (Kanto Chemical Co., Inc.; 99.5%) were adopted as a solvent and a base catalyst in alkanoyl substitutions of HPC, respectively. As three kinds of alkanoyl chlorides, acetyl chloride (Tokyo Chemical Industry Co., Ltd., Tokyo, Japan; 98.0%), propionyl chloride (Tokyo Chemical Industry Co., Ltd.; 98.0%), and butyryl chloride (Tokyo Chemical Industry Co., Ltd.; 98.0%) were used without further purification. Ultrapure water, used as a poor solvent in reprecipitation, was prepared using a water purification system (Direct-Q UV 5, Merck Millipore, Burlington, MA, USA).

### 3.2. Synthesis and Characterization of HPC Ester Derivatives

Figure 1 shows the chemical structure of pristine HPC and its ester derivatives possessing alkanoyl side chains. Three kinds of HPC derivatives were synthesized by esterification of hydroxy groups in pristine HPC with alkanoyl chlorides such as acetyl chloride, propionyl chloride, and butyryl chloride according to our previous report [29]. For instance, the typical synthesis procedure of an HPC ester derivative possessing acetyl moieties in the side chains (Figure 1, HPC-Et) is described as follows.

In a 300 mL three-neck round-bottom flask with nitrogen substitution, 6.00 g of dried HPC (1.00 eq. to the number of HPC monomer units in 6.00 g of HPC) was completely dissolved in 90.0 mL of anhydrous acetone by stirring. Subsequently, 7.57 g of anhydrous pyridine (6.30 eq.) was added as a base catalyst to the solution at room temperature. After continuously stirring for 30 min, the solution was refluxed at 55 °C, followed by the dropwise addition of 7.16 g of acetyl chloride (6.00 eq.). After the reaction proceeded for 20 h, the supernatant of the reaction solution was dropped into ~1.00 L of ultrapure water to yield a white sticky product. This product was purified by three cycles of reprecipitation from acetone to ultrapure water. Finally, the purified product was dried in vacuo at room temperature for at least 48 h. Likewise, we prepared the other HPC derivatives, HPC-Pr and HPC-Bu, in the same manner as pristine HPC, reacting with 8.44 g of propionyl chloride (6.00 eq.) and 9.72 g of butyryl chloride (6.00 eq.) instead of acetyl chloride, respectively.

In order to confirm the reactivity of hydroxy groups in HPC with alkanoyl chlorides, FT-IR and ^1^H-NMR spectra were measured for pristine HPC and its derivatives. FT-IR spectra were acquired using an FT-IR spectrometer (FTIR-4700, JASCO, Tokyo, Japan) equipped with an attenuated total reflection (ATR) unit of diamond prism (ATR PRO ONE, JASCO), and ^1^H-NMR spectra in CDCl_3_ with an internal standard of tetramethylsilane were recorded on an NMR spectrometer (ECZ 400, JEOL, Tokyo, Japan). The values of *M*_n_ and *M*_w_ of HPC derivatives were determined by an SEC system (HLC-8220 GPC, TOSOH, Tokyo, Japan) combined with a refractive index detector. The SEC measurements were carried out at 40 °C by flowing tetrahydrofuran (Kanto Kagaku Co., Ltd., Tokyo, Japan; 99.5%) as an eluent at a flow rate of 0.35 mL/min. Lastly, the values of *M*_n_ and *M*_w_ were evaluated by calibrating using the polystyrene standards.

### 3.3. Optical Measurements

In order to evaluate the reflection properties, the HPC derivatives were enclosed between two glass substrates to fabricate the CLC cells. The internal gap distance (*d*_g_) between the glass substrates was adjusted by polytetrafluoroethylene film spacers with a thickness of ~0.20 mm. Transmission spectra of the CLC cell were measured using a compact charge-coupled device (CCD) spectrometer (USB2000+, Ocean Optics, Orlando, FL, USA) equipped with a tungsten halogen light source (Ocean Optics, HL-2000) for the probing white light. The collinearly transmitted light from the CLC cell was focused through two pieces of achromatic doublet lenses and collected into the core of an optical fiber connected with the CCD spectrometer. The temperature of the CLC cell was precisely controlled using a temperature controller (HS1, Mettler-Toledo, Columbus, OH, USA) equipped with a hot stage (HS82, Mettler-Toledo) [25]. Polarized optical microscope (POM) images were taken with a CCD camera (EO-5012, Edmund, Barrington, NJ, USA) equipped on the microscope (IX71, OLYMPUS, Tokyo, Japan).

### 3.4. X-ray Diffraction Measurements

Wide-angle X-ray diffraction (WAXD) measurements of HPC derivatives were performed on a benchtop X-ray diffractometer (MiniFlex 600, Rigaku, Tokyo, Japan) with Cu-Kα radiation equipped with a temperature controller (HPC-200, Rigaku). Before measurements, the HPC derivatives were heated at ~130 °C on a hot stage for 30 min in order to eliminate any remaining trace of molecular orientation caused by placing the samples on substrates. WAXD measurements in the reflection mode were conducted at the temperature where the HPC ester derivatives showed a reflection peak at a wavelength of 405 nm. The scanning range of 2*θ* was adjusted between 2° and 30°.

### 3.5. Rheological Measurements

The rheological properties of storage (*G*′) and loss (*G*″) moduli were assessed using a stress-controlled rheometer (MCR 102, Anton Paar, Graz, Austria) equipped with a stainless-steel parallel plate with a diameter of 8 or 25 mm, which was used differently depending on the elastic moduli of the HPC ester derivatives at each temperature. The temperature was precisely controlled by a convection temperature device (CTD450, Anton Paar). The gap distance (*d*_g_), which corresponds to the film thickness of HPC derivatives between parallel plates and the sample stage in the rheometer, was adjusted to 0.20, 0.25, 0.30, 0.75, and 1.00 mm in order to elucidate the effect of *d*_g_ on the rheological properties. The measurements were performed at 100 °C, at a strain amplitude of 0.7%, and in a range of angular frequency (*ω*) between 10^−1^ and 10^2^ rad/s. Before measurements, the HPC derivatives were pre-sheared at a shear rate of 1 s^−1^ for 5 min and then left to stand for 2 min to ensure uniform CLC orientation [25]. This interval was sufficiently long to relax and orient the CLCs of HPC derivatives, whose helical axis was tilted by pre-shearing (Appendix A).

In order to construct the master curves of HPC-Et, HPC-Pr, and HPC-Bu, the values of *G*′ and *G*″ were measured in the range of *ω* between 10^−1^ and 10^2^ rad/s, at temperatures ranging from −30 to 112 °C. At this time, the *d*_g_ value was adjusted from 0.33 to 0.60 mm. In this *d*_g_ range, the values of *G*′ and *G*″ were identical, as will be mentioned below. Liquid nitrogen was used as the cooling source for the low-temperature measurements. The strain amplitude was adjusted in the regime of linear viscoelasticity. The experimental results of *G*′ and *G*″ were shifted vertically and horizontally in logarithmic scales according to the time–temperature superposition (TTS) principle with the reference temperatures (*T*_r_), where each HPC ester derivative showed a reflection at 405 nm. The horizontal shift factors (*a*_T_) were determined by shifting the density-independent loss tangent (tan *δ*), corresponding to *G*″/*G*′, while the vertical shift factors (*b*_T_) were determined by shifting *G*′ and *G*″ vertically. The activation energies for relaxation (*E*_a_) were calculated from the temperature dependence of the *a*_T_ value.

## 4. Conclusions

In this study, we investigated the linear rheological phenomena of the HPC derivatives possessing acetyl, propionyl, or butyryl side chains, which exhibited a thermotropic CLC phase with visible reflection. The complete esterification of HPC by alkanoyl chlorides was confirmed by both ^1^H-NMR and FT-IR spectra. The reflection peak wavelengths were red-shifted with the increase in temperatures or the length of alkanoyl groups. When their temperatures were adjusted to reflect a blue light at 405 nm, they exhibited almost the same *ω* dependences of *G*′ and *G*″. This is because the derivatives have almost the same CLC structures. Furthermore, the relaxation process due to the tilt of the helical axis of CLC is probably dominant in the low *ω* range, specifically below 10^2^ rad/s, and *G*′ and *G*″ are not affected by the difference in the chemical structures of HPC derivatives. This discussion is also supported by the gap dependence of *G*′ and *G*″, which indicates that the increase in *G*′ and *G*″, caused by the enhancement of planer orientation, is more apparent at the lower *ω* side. Since the activation energy of the relaxation process tends to decrease as the length of alkanoyl groups increases, the derivative that has longer alkanoyl side chains may be more advantageous in improving the orientation of CLCs. These results will contribute to our further understanding of the orientation behavior of CLCs. We are currently trying to elucidate the effect of the activation energies on the orientation behavior of CLCs by optical measurements, which will be reported in our upcoming articles.

## Figures and Tables

**Figure 1 ijms-24-04269-f001:**
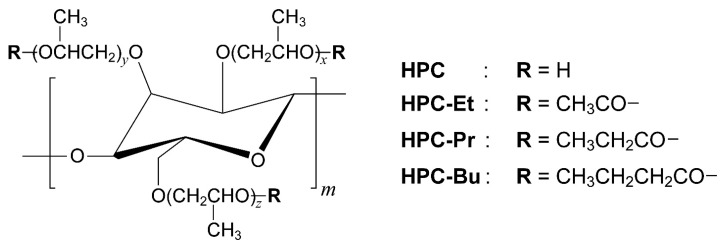
Chemical structures of HPC and its derivatives possessing alkanoyl side chains. The terminal hydroxy groups of pristine HPC were fully substituted with acetyl (ethanoyl), propionyl, or butyryl groups. All HPC derivatives of HPC-Et, HPC-Pr, and HPC-Bu showed thermotropic CLC phase with visible reflection as heating above ~110, ~95, and ~60 °C, respectively.

**Figure 2 ijms-24-04269-f002:**
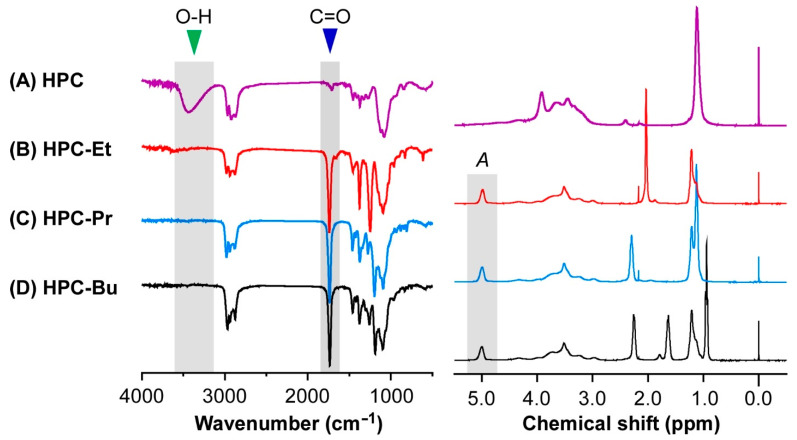
Attenuated total reflection (ATR) FT-IR spectra (left side) and ^1^H-NMR spectra (right side) of HPC (A; purple lines), HPC-Et (B; red lines), HPC-Pr (C; blue lines), and HPC-Bu (D; black lines). Two peaks at 1700 cm^−1^ (indicated by blue triangle) and 3100 cm^−1^ to 3600 cm^−1^ in the ATR FT-IR spectra (indicated by green triangle) are assigned to C=O stretching vibration and O-H stretching vibration, respectively. *A* is the integrated values of the peaks appearing at 5.00 ppm in the ^1^H-NMR spectra, meaning the assignment of the methine protons of the esterified hydroxypropyl groups.

**Figure 3 ijms-24-04269-f003:**
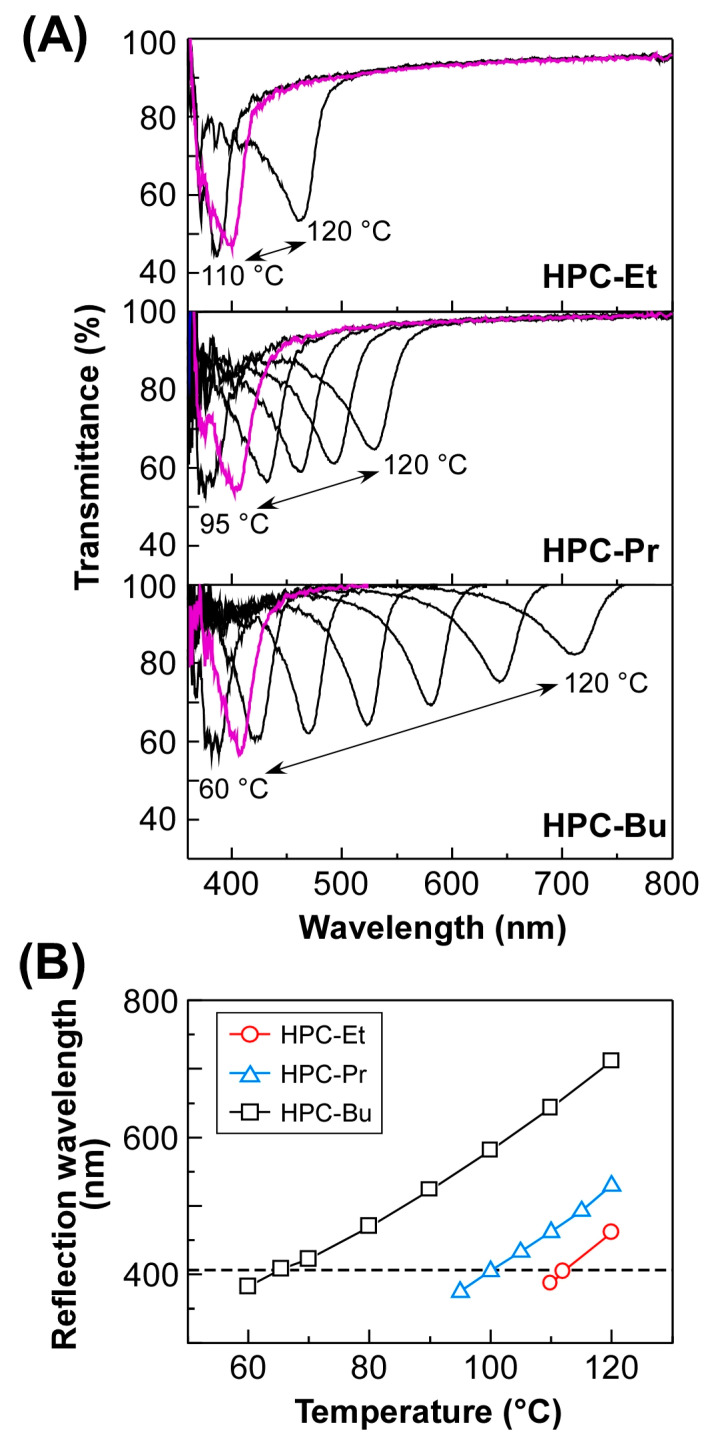
Reflection properties of a series of HPC derivatives upon changing the temperature. (**A**) Changes in the transmission spectra of HPC-Et (**upper panel**), HPC-Pr (**middle panel**), and HPC-Bu (**lower panel**) in the temperature range between 60 and 120 °C. Each spectrum was recorded at the intervals of 10 °C for HPC-Et, 5 °C for HPC-Pr, and 10 °C for HPC-Bu (black lines). A reflection peak appeared at 405 nm by heating at 112 °C for HPC-Et, 100 °C for HPC-Pr, and 66 °C for HPC-Bu (purple lines). (**B**) Temperature dependences of reflection peak wavelengths of HPC-Et (red circles), HPC-Pr (blue triangles), and HPC-Bu (black squares). The horizontal dotted line corresponds to the reflection peak wavelength of 405 nm.

**Figure 4 ijms-24-04269-f004:**
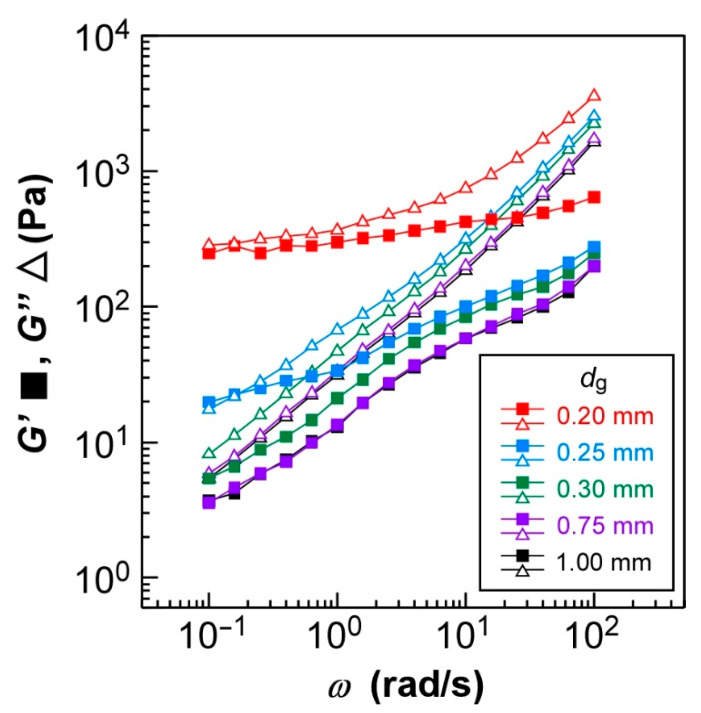
Effect of gap distance (*d*_g_) of HPC-Pr on its rheological properties. Changes in the storage modulus (*G*′, closed squares) and loss modulus (*G*″, open triangles) of HPC-Pr as a function of angular frequency (*ω*). The rheological measurements were performed at 100 °C, where a reflection peak of CLC phase appeared at 405 nm (Figure 3), and the *d*_g_ values were adjusted to 0.20 (red symbols), 0.25 (blue symbols), 0.30 (green symbols), 0.75 (purple symbols), and 1.00 mm (black symbols). Notice that the profiles of *G*′ and *G*″ were almost overlapped at the *d*_g_ values of 0.75 and 1.00 mm.

**Figure 5 ijms-24-04269-f005:**
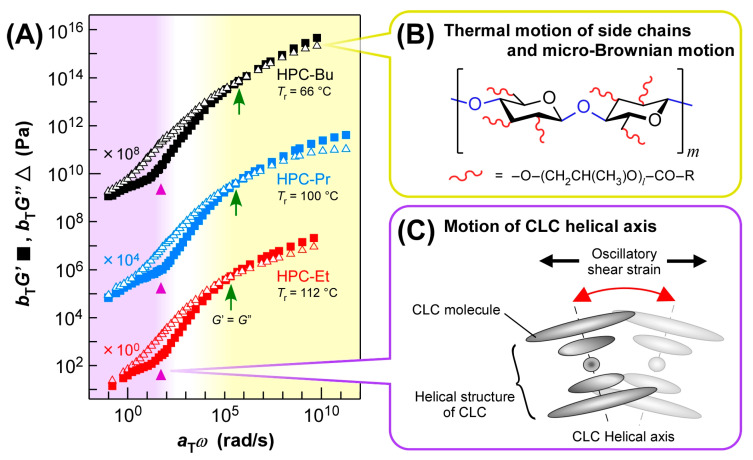
(**A**) Master curves of HPC-Et (red plots, lower profiles), HPC-Pr (blue plots, middle profiles), and HPC-Bu (black plots, upper profiles) with reference temperatures (*T*_r_) at which these HPC derivatives showed a reflection peak at 405 nm. All the master curves were prepared by time–temperature superposition (TTS) principle. The values of *a*_T_ and *b*_T_ are horizontal and vertical shift factors, respectively. Note that the master curves of HPC-Pr and HPC-Bu are shifted in the vertical direction of profiles by increasing both *b*_T_*G*′and *b*_T_*G*″ values for the reader’s clarity. The green arrows stand for the intersections of *G*′ and *G*″, and the purple triangles are the inflection points of *G*′. (**B**) Schematic illustration of the dominant relaxation process in the higher *ω* region (the region highlighted in yellow in Figure 5A). In this illustration, *l* means the number of repeating propyleneoxy units of HPC. (**C**) Schematic illustration of the dominant relaxation process in the lower *ω* region (the region highlighted in purple in Figure 5A).

**Figure 6 ijms-24-04269-f006:**
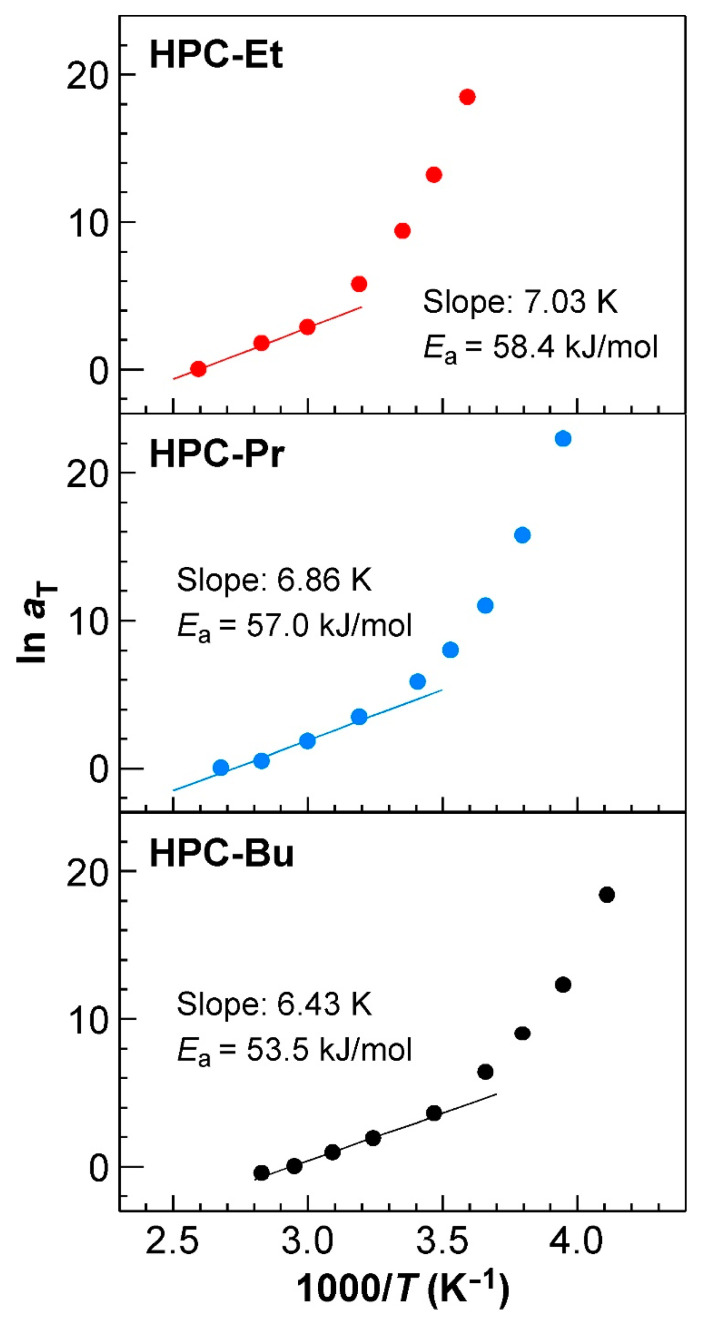
Arrhenius-type plots of the *a*_T_ values of HPC-Et (**upper panel**), HPC-Pr (**middle panel**), and HPC-Bu (**lower panel**). The solid lines represent their regression lines of *a*_T_ in the high-temperature region. The activation energies (*E*_a_) of the HPC derivatives were calculated from the slopes of regression lines, as noted in the figures.

**Table 1 ijms-24-04269-t001:** Characterization of pristine HPC and its ester derivatives possessing alkanoyl side chains with the different lengths.

Sample	*DE*	*M*_w_ (×10^4^)	*M*_w_/*M*_n_
Pristine HPC	-	4.45	1.93
HPC-Et	3.00	4.79	1.79
HPC-Pr	2.98	5.97	1.73
HPC-Bu	2.98	6.77	2.12

**Table 2 ijms-24-04269-t002:** Experimental results of WAXD measurements of HPC-Et, HPC-Pr, and HPC-Bu at the temperature where the HPC derivatives showed a reflection peak at 405 nm. The distance (*d*) and the twisting angle (*φ*) between two adjacent layers, and the number of layers in one pitch (*N*_Layer_) of each sample were determined using 2*θ* values obtained from the WAXD measurements.

Sample	2*θ* (°)	*d* (nm)	*φ* (°)	*N* _Layer_
HPC-Et	7.17	1.23	1.60	224
HPC-Pr	6.84	1.29	1.68	214
HPC-Bu	6.65	1.33	1.73	208

## Data Availability

Data is contained within the article.

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
