# Peer review of "Dominant Factors Affecting Rheological Properties of Cellulose Derivatives Forming Thermotropic Cholesteric Liquid Crystals with Visible Reflection"

_ijms, 2023, doi:10.3390/ijms24054269_

Round 1
Reviewer 1 Report
Please see the attached pdf for comments.

Author Response
We want you to read the PDF file of our "Point-by-point responses" document uploaded on this web site.

Reviewer 2 Report
The manuscript “Dominant Factors Affecting Rheological Properties of Cellulose Derivatives Forming Thermotropic Cholesteric Liquid Crystals with Visible Reflection” was reported on the synthesis work of Hydroxypropyl cellulose (HPC) derivatives with alkanoyl side chains. However, there is a lack of discussion on the structure-property relationship that affects the rheological properties of cellulose derivatives and finally induces thermotropic cholesteric liquid crystals. No chemical interpretation was discussed to confirm the elucidation of each derivative.
- Abstract:
- The authors should highlight the benefit of the study, such as advantages, function, readiness to be used, etc.
- The qualitative data related to chemical characterization should be included.
- Introduction:
- Please elaborate more on the background of the study, current progress, and future application for the synthesized compounds.
- Even though the reported compounds are considered new, the author should add an example from the previous journal in the introduction section to show the development/trend of the improvement of the study.
- Experimental & Technical Issues:
- How do the authors monitor the reaction progress?
- There are a lot of mistakes in citing the Figure number in the text. Some of the figures are not found in the manuscript and supplementary files. The authors MUST arrange it properly to avoid confusion for the reader.
- No chemical analysis on NMR and FTIR spectral discussion in the manuscript.
- No POM analysis was found, which contradicts the manuscript's title. Since the other characterization related to CLCs was studied, however, the type of thermotropic cholesteric LCs was not included. This is the MUST for the research study.
- Conclusion:
- Please includes the qualitative data (chemical characterization)
- Describe the limitations of the study.
· References: Some of the references provided are articles with more than 5 years. Please cite recent articles to support the project conducted is up-to-date research.
· Language Style: The quality of the English language of the whole does not meet the standard necessary for a scientific article. I would therefore recommend that the authors should look for the support of a native English speaker.
Author Response

(The authors gave the same response as above.)

Reviewer 3 Report
The article is well written, and the manuscript was easy to follow. The number of figures is adequate and reveals an extensive experimental work done. The introduction explains in a concise yet precise manner many of the conclusions so far drawn on this subject.
The discussion focuses on the experimental facts and applied well know principles. The discussion of the results is, in the majority of the text, very clear, and only minor explanations seemed to be needed.
The conclusions are pertinent and sound.
The number of references is good and allows the reader, if not experienced in the area of CLC derived from cellulose and rheological studies, to follow what is being presented. Although I will suggest some small additions to some sentences.
Overall, I think this is a very complete and interesting work, and I suggest minor corrections. Moreover, I believe this will be an important study that will be used by future researchers.
In general terms, some revision regarding the number of figures and where they are located should be done.
Examples:
1- The authors present figure 5 first in the text, and it will appear much below.
2-The authors also introduce figure 1 in the main text as the chemical structures of HPC derivatives, although such figure does not have chemical structures. (Pg 2 line 109)
3- The authors introduce in the text figure S2 before S1 and the derivatization at the end of the supplementary information. In general, I think the text will be easy to follow if all the figures are in order and in line with what is presented in the text and in the supplementary information.
Some revisions of the figure’s captions should also be made. For instance, the black lines in figure 1 are not introduced.
Line 71: The authors stated that “HPC derivatives can be regarded as the most suitable… “ I think there are other systems derived from cellulose that can also be used. In my opinion, this sentence should be reformulated since other cellulosic systems are able to be used.
Section 2.2: I think it should be clear at the beginning of this study that the authors intend to determine the temperature at which the HPC derivatives reflected 405 nm. My comment arises from analyzing figure 1, where different temperatures and a distinct number of data are presented for the three systems.
After and while reading the manuscript, one can find the transition temperature values from CLC to isotropic phase, but only much further. So perhaps some further explanation should be added to this initial part.
The authors mentioned the reversibility of the systems; however, no spectra are shown, even on supplementary information. What is the standard deviation obtained for this analysis? Standard deviation should be added at least to figure 1B.
Pg 9, line 320, the authors stated, “As the dg increased, the baselines of the transmission spectrum were gradually lowered, accompanied by broadening in the spectral widths of reflection peaks.” Indeed, figure S4 shows these effects, and scattering might be responsible for these differences, but figure S4 also shows a change in the maximum reflection peak, which the authors do not explain. Are these differences within the standard deviation observed in the reversibility measurements?
The YY scale in figure S5 should be narrow so one can see the change in the maximum wavelength in an easier manner. Perhaps a scale of time should also be added to improve reading this figure.
Some small revision of the language mainly the use of some articles should be done.
Minor corrections:
Line 29 remove of after those
Line 33 remove as after called
Line 39 remove the after water at
Line 44 remove In other word to This liquid crystal phase…
Line 46 add some references after HPC derivatives
Line 58 it seems that the word all is missing before HPC derivatives
Line 167 after chains add some references
Line 423 correct deference to difference
Line 499 add one reference after WLF method
Pg 13 1st paragraph. Add a reference to the Arrhenius plots vs at values and bring here the presentation of the supplementary information regarding the calculation of Ea
Author Response

(The authors gave the same response as above.)

Round 2
Reviewer 2 Report
Sorry for the late response. Based on the correction that has been submitted, the manuscript can be accepted after a minor correction (grammar checking).
Author Response

(The authors gave the same response as above.)
